# Cellular HIV Reservoirs and Viral Rebound from the Lymphoid Compartments of 4′-Ethynyl-2-Fluoro-2′-Deoxyadenosine (EFdA)-Suppressed Humanized Mice

**DOI:** 10.3390/v11030256

**Published:** 2019-03-13

**Authors:** Ekaterina Maidji, Mary E. Moreno, Jose M. Rivera, Pheroze Joshi, Sofiya A. Galkina, Galina Kosikova, Ma Somsouk, Cheryl A. Stoddart

**Affiliations:** 1Division of Experimental Medicine, Department of Medicine, Zuckerberg San Francisco General Hospital, University of California, San Francisco, CA 94110, USA; ekaterina.maidji@ucsf.edu (E.M.); marybeth.moreno@ucsf.edu (M.E.M.); jose.rivera@ucsf.edu (J.M.R.); pheroze.joshi@ucsf.edu (P.J.); sofiya.galkina@ucsf.edu (S.A.G.); galina.kosikova@ucsf.edu (G.K.); 2Division of Gastroenterology, Department of Medicine, Zuckerberg San Francisco General Hospital, University of California, San Francisco, CA 94110, USA; ma.somsouk@ucsf.edu

**Keywords:** HIV, infection, cellular reservoir, antiretroviral therapy, macrophage, DC-SIGN, RNAscope

## Abstract

Although antiretroviral therapy (ART) greatly suppresses HIV replication, lymphoid tissues remain a sanctuary site where the virus may replicate. Tracking the earliest steps of HIV spread from these cellular reservoirs after drug cessation is pivotal for elucidating how infection can be prevented. In this study, we developed an in vivo model of HIV persistence in which viral replication in the lymphoid compartments of humanized mice was inhibited by the HIV reverse transcriptase inhibitor 4′-ethynyl-2-fluoro-2′-deoxyadenosine (EFdA) to very low levels, which recapitulated ART-suppression in HIV-infected individuals. Using a combination of RNAscope in situ hybridization (ISH) and immunohistochemistry (IHC), we quantitatively investigated the distribution of HIV in the lymphoid tissues of humanized mice during active infection, EFdA suppression, and after drug cessation. The lymphoid compartments of EFdA-suppressed humanized mice harbored very rare transcription/translation-competent HIV reservoirs that enable viral rebound. Our data provided the visualization and direct measurement of the early steps of HIV reservoir expansion within anatomically intact lymphoid tissues soon after EFdA cessation and suggest a strategy to enhance therapeutic approaches aimed at eliminating the HIV reservoir.

## 1. Introduction

Despite the extraordinary success of antiretroviral therapy in suppressing viremia in HIV-infected individuals [1,2,3], ART cannot eliminate all persistently infected cells enabling a stable viral reservoir that remains in the body, and HIV re-establishes active infection after ART is discontinued [4,5,6,7,8,9]. HIV most commonly enters the body through mucosal surfaces and then disseminates throughout the lymphoid tissues [10,11,12,13,14], which serve as sanctuary sites where HIV may replicate at low levels despite potent ART and undetectable viremia [3,7]. It is believed that cell-to-cell spread is a crucial factor enabling virus replication during ART [15,16]. The mechanisms involved in the maintenance and persistence of the HIV reservoir could be multi-factorial and may include insufficient penetration of antiviral drugs into tissues where virus persistently replicates [3,5], intrinsic stability, and periodic proliferation of latently infected resting CD4^+^ T cells bearing an infectious virus [4,6,17,18,19,20], and the survival of the infectious virus for extended periods in macrophages [21]. Given the complexity of HIV persistence and the variety of current curative strategies for eliminating or suppressing the viral reservoir, identification of persistently infected cells and their anatomical location as well as study of the effect of tissue environment on viral gene expression and replication in vivo are critical for understanding HIV pathogenesis and for developing effective HIV eradication regimens.

Considering the practical and ethical issues that surround the collection of tissues from ART-suppressed HIV-infected individuals, humanized mice represent invaluable systems to study the persistent HIV reservoir. After introduction of the SCID-hu Thy/Liv in 1990 [22], several different humanized mouse models were developed using severe combined immuno-deficient (*scid*) mice and widely used for study of HIV pathogenesis and drug development [23,24,25]. More recently, humanized mouse models have been developed that employ NOD-*scid* mice homozygous for a deletion of the IL-2R γ-chain (NOD-*scid* IL-2Rγ^−^/^−^, also called NSG) [26]. Implantation of these mice with human thymus and autologous liver under the kidney capsule generate T cell-only mice (ToM) that lack human B cells, monocytes, macrophages, and dendritic cells [27]. When NSG mice are co-implanted with human thymus and autologous liver under the kidney capsule and receiving autologous human CD34^+^ hematopoietic cells, this generates the bone marrow-liver-thymus (BLT) mouse model [28,29]. BLT mice are reconstituted systemically with virtually all human hematopoietic cell types, including T cells, B cells, monocytes, NK cells, macrophages, and dendritic cells. An attractive novel model, myeloid–only mice (MoM), was recently suggested for an in vivo study of HIV replication in macrophages [30]. The MoM model was generated by implantation of NOD-*scid* mice with human CD34^+^ hematopoietic cells. These mice were distinctive because of their reconstitution with human B cells and myeloid cells but a lack of T cells. Both ToM and MoM models allow the study of viral pathogenesis in T cells and macrophages independently.

In this study, we investigated the role of interactions between macrophages and T cells in HIV pathogenesis using NSG-BLT mice that were recently proposed as a valuable model to evaluate novel approaches for HIV eradication in tissues [31]. As we reported earlier, HIV viremia in NSG-BLT mice inoculated with HIV_JR-CSF_ could be fully suppressed after two weeks of treatment with the highly potent HIV reverse transcriptase translocation inhibitor EFdA [32]. EFdA has a prolonged intracellular half-life in human and rhesus macaque peripheral blood cells, excellent tissue penetration, and robust antiviral activity of 7 to 10 days’ duration [33,34]. We used data from these published reports to develop an in vivo model of HIV persistence in which viral replication in the lymphoid compartments of humanized mice was inhibited by EFdA to very low levels. This recapitulates ART-suppression in HIV-infected individuals. We then applied a combination of immunohistochemistry and an ultrasensitive, semi-quantitative RNAscope in situ hybridization to characterize cells quantitatively in the lymphoid compartments of HIV-infected humanized mice wherein the virus resides during (1) active infection, (2) fully suppressive EFdA treatment, and (3) after drug cessation. Our data allowed visualization and measurement of the early steps of HIV reservoir expansion within anatomically intact lymphoid compartments soon after EFdA cessation and suggested a strategy to prolong viral control and reduce the number of HIV-infected cells.

## 2. Materials and Methods

### 2.1. Ethics Statement

This study was carried out in strict accordance with the recommendations in the Guide for the Care and Use of Laboratory Animals of the National Institutes of Health. The UCSF Institutional Animal Care and Use Committee approved all animal protocols (AN176275-01A, approval date: 25 September 2018).

### 2.2. NSG-BLT Mice

NSG-BLT mice were generated, as described by Melkus et al. [29] using 12-week-old female NSG mice (NOD.Cg-*Prkdc^scid^Il2rg^tm1Wjl^*/SzJ, Jackson Laboratories). Autologous CD34^+^ hematopoietic stem cells (HSC) were purified by collagenase dispersion, isolation of mononuclear cells by centrifugation over Lymphoprep 1.077 (Axis-Shield), and magnetic bead selection for CD34^+^ cells (StemCell). HSC were cryopreserved in 90% fetal bovine serum/10% dimethylsulfoxide and stored in liquid nitrogen until injection into mice three weeks after Thy/Liv implantation. The mice were not irradiated prior to the cell injection because we found that this conditioning step is not necessary for robust human leukocyte reconstitution.

### 2.3. Flow Cytometry

Human leukocyte reconstitution was assessed by flow cytometry using Trucount (BD Biosciences, San Jose, CA, USA) enumeration to calculate the absolute number of human B cells, CD4^+^ and CD8^+^ T cells, monocytes, NK cells, and neutrophils per μL of blood [35]. Anti-human CD45 (clone H130, BD Biosciences, San Jose, CA, USA) and anti-mouse CD45 (clone 30-F11, BD Biosciences, San Jose, CA, USA) antibodies were used to differentiate mouse from human leukocytes. Human CD45^+^ cells were phenotyped using antibodies specific for human CD3 (clone UCHT1, Beckman Coulter, Brea, CA, USA), CD4 (clone RPA-T4, Biolegend, San Diego, CA, USA), CD14 (clone TüK4, Invitrogen, Carlsbad, CA, USA), and CD19 (clone HIB19, Biolegend, San Diego, CA, USA). Data analysis was performed using FlowJo software (v.9.9.4, FlowJo LLC, Ashland, OR, USA).

### 2.4. HIV-1 Inoculation and EFdA Treatment

An infectious virus stock was generated in HEK 293T cells by lipofectamine 2000 transfection with pYK-JRCSF [36,37,38] plasmid obtained from the NIH AIDS Reagent Program. The virus titer was determined in phytohemagglutinin (PHA)-stimulated peripheral blood mononuclear cells (PBMC) by serial dilution and assessment of supernatant HIV p24 by ELISA (PerkinElmer Life Sciences, Waltham, MA, USA) after 7 days, and 50% tissue culture infectious doses (TCID_50_) were calculated using the Reed-Muench method. At 8 weeks after CD34^+^ cell injection, 2.5 × 10^4^ TCID_50_ of HIV-1_JR-CSF_ was inoculated intraperitoneally (i.p.). Beginning 2 weeks after HIV inoculation, mice were treated for 4 weeks with 10 mg/kg/day EFdA by once-daily oral gavage. Peripheral blood, Thy/Liv implants, spleens, and lungs were collected from euthanized mice 6 and 8 weeks after virus inoculation. Four mice in the same cohort were not inoculated and served as negative controls.

The level of viral RNA copies per 100 µL plasma was analyzed by the Abbott RealTime HIV-1 Viral Load (PCR) assay. Single-cell suspensions were prepared by gently dispersing spleens through a 100-µm cell strainer and flushing with RPMI 1640 medium. Cells were stained with Muse count and viability reagents and counted with the Muse cell analyzer (Millipore, Burlington, MA, USA). Infectivity of the spleen cells was assessed in an infectious unit per million (IUPM) cells assay by serial dilution, co-cultivation with PHA-stimulated PBMC, supernatant p24 determination by ELISA at 7 days, and Reed-Muench calculation [39].

### 2.5. RNAscope in situ Hybridization and Immunohistochemistry

Mouse spleens, lungs, and Thy/Liv implants were removed immediately after euthanasia, fixed in formalin for 24 h, and embedded in paraffin. ISH was performed using RNAscope, which is a novel next-generation technique developed by Advanced Cell Diagnostics. This technology utilizes a unique “double Z” probe design that greatly increases the signal-to-noise ratio and can visualize RNA transcripts at the single-molecule level. RNAscope ISH for HIV RNA and DNA was performed using the 2.5 HD Reagent RED kit (Advanced Cell Diagnostics, Newark, CA, USA), according to the manufacturer’s instructions. For detection of HIV DNA, tissue sections were pretreated with 25 µg/mL ribonuclease A (Fisher Scientific, Hampton, NH, USA) and 25 U/mL ribonuclease T_1_ (Roche Diagnostic, Risch-Rotkreuz, Switzerland) in Tris-buffered saline for 30 min at 37 °C prior to target hybridization. Target DNA in situ hybridization including a 15-min denaturing step at 60 °C was performed as described [40]. HIV RNA was detected using an HIV-1-*gag-pol* anti-sense probe (Advanced Cell Diagnostics) that targets *gag-pol* coding sequence region 587–4601. HIV DNA was detected using the HIV-1-Clade B-sense probe (Advanced Cell Diagnostics) that targets the integrated HIV DNA noncoding sequence regions 854–8291 (*gag*, *pol*, *vif*, *vpr*, *tat*, *rev*, *vpu*, *env*) of the HIV clade B NL4-3 isolate. To confirm specificity of ISH, we used spleen sections from control NSG mice as a negative control (Appendix A). Human peptidyl-prolyl cis-trans isomerase B encoded by the *PPIB* gene was detected with the Hs-*PPIB* probe in the HeLa cell line control (both from Advanced Cell Diagnostics, Newark, CA, USA) and served as an RNAscope positive control (Appendix A). The RNAscope assay was followed by standard IHC for human CD3, CD163, CD68, or HIV p24. Primary antibodies were mouse mAb anti-HIV-1 p24 (183-H12-5C) from the AIDS Reagent Program and anti-human CD3 (clone F7.2.38, Diagnostic BioSystems, Pleasanton, CA, USA), CD163, (Leica Biosystem, Wetzlar, Germany), CD68 (clone KP-1, Agilent, Santa Clara, CA, USA), rabbit mAb anti-human CD163 (EPR14643-36, Abcam, Cambridge, UK), CD3 (SP7, Abcam, Cambridge, UK), and rabbit polyclonal Ab anti-human DC-SIGN (Abcam, Cambridge, UK). The secondary antibody, ImmPRESS polymeric HRP-linked horse anti-mouse IgG, was detected using ImmPACT SG HRP substrate (both from Vector Laboratories Inc., Burlingame, CA, USA). HIV RNA and integrated viral DNA were visualized by alkaline phosphatase (AP) using the Fast-Red substrate. In some experiments, fluorescence detection of Fast Red using a far-red filter was used. Nuclei were counterstained with hematoxylin QS (Vector Laboratories Inc., Burlingame, CA, USA). For fluorescent labeling, the RNAscope multiplex fluorescent kit V2 (Advanced Cell Diagnostics, Newark, CA, USA) was used with PerkinElmer TSA plus system fluorophores. The RNAscope multiplex fluorescent assay combined with immunofluorescence was performed, according to the manufacturer’s instructions from Advanced Cell Diagnostics. Tissue sections were analyzed with a Leica DM6000 B microscope equipped with a Leica DFC 500 camera, and images were acquired using LAS v4.3 software (Leica, Wetzlar, Germany).

### 2.6. Data Analysis

The numbers of HIV RNA^+^ or HIV DNA^+^ cells in various tissue samples were assessed using 3 or 4 cross-sections taken at different levels of the tissue block. In addition, 30 to 50 high-power (63×) microscope fields of the regions containing human cells were randomly chosen from 5 to 10 different areas of the tissue section and captured digitally with the system described above. Image acquisition parameters were kept constant throughout each experiment. Each captured field included an area of approximately 0.04 mm^2^, and ImageJ software was used to quantify the number of positive cells. The numbers of HIV RNA^+^ or HIV DNA^+^ cells were presented as cells per mm^2^. When comparisons were performed between different organs (spleen, lung, and Thy/Liv implants), the number of HIV RNA^+^ cells was calculated per 10^5^ cells [40]. In brief, conventional bright-field RGB color images were split by ImageJ. The red component (greater contrast nuclei staining) was selected [41] and used to assess the total number of cells per image. Nuclei were segmented by thresholding the red channel. The segmentation was then watershed, and the segmented object was counted. The number of HIV RNA^+^ cells per image was divided by the estimated number of cells determined by the nuclei segmentation in that image and multiplied by 10^5^ [40].

Statistical analysis was performed by the nonparametric Mann–Whitney U-test. A *P* value of <0.05 was considered statistically significant.

## 3. Results

### 3.1. Human T Cells and Macrophages Constitute a Substantial Population of HIV-Susceptible Cells in the Lymphoid Compartments of NSG-BLT Mice

Since, in HIV-infected individuals, HIV RNA was detected at higher frequencies in primary (tonsil and bone marrow) [42] and secondary lymphoid tissues, such as gut-associated lymphoid tissue (GALT) [43,44] and lymph nodes [45], our experiments focused on the investigation of the cellular reservoir of HIV in the lymphoid compartments of NSG-BLT mice. To document the superior human leukocyte engraftment and susceptibility to HIV infection in the lymphoid compartments of NSG-BLT mice, we employed a combination of state-of-the-art RNAscope ISH technology and IHC. NSG-BLT mice were inoculated i.p. with HIV-1_JR-CSF_ and, 2 weeks later, Thy/Liv implants and mouse lungs and spleens were collected for tissue analysis. At this stage of infection, the lymphoid white pulp of the mouse spleens displayed a high level of human CD3 cell engraftment, and many of CD3^+^ cells contained HIV RNA (Figure 1A). RNAscope ISH is an exceptionally sensitive method that allows the visualization of individual RNA molecules as intensely staining discrete puncta [46]. In cells expressing a high level of HIV RNA, individual puncta fuse together and appear as dense staining of the entire cell. Therefore, we were able to distinguish cells containing a high level of HIV RNA (dense fuchsia staining in the nucleus and cytoplasm) from those with a low level of HIV RNA (discrete fuchsia puncta in the nucleus and cytoplasm) (Figure 1A, inset). Although the majority of HIV RNA^+^ cells were hCD3^+^ T cells, human CD68^+^CD163^+^ macrophages containing single fuchsia puncta in the nucleus were detected occasionally in the marginal zone (Figure 1B, inset). Although the RNAscope *gag-pol* probe that was used in the assay is optimally designed to detect HIV RNA, it may also detect the sense strand of integrated HIV DNA. Consequently, discrete fuchsia puncta identified in the nucleus of human CD68^+^CD163^+^ macrophages may indicate integrated HIV DNA and be a sign of ongoing virus replication. Notably, the distribution of HIV RNA^+^ cells in the spleen white pulp and the Thy/Liv implants was similar, and the Thy/Liv implants also revealed numerous HIV RNA^+^ T cells (Figure 1C). Human CD68^+^CD163^+^ macrophages intercalated between T cells occasionally contained single fuchsia puncta in the nucleus and cytoplasm (Figure 1D, inset), which suggests HIV replication in macrophages.

In the lung of NSG-BLT mice, we consistently detected abundant bronchus-associated lymphoid tissue (BALT) as a site of lymphoid organogenesis (Appendix A). Although the frequency of BALT in mice is very variable [47], we could sometimes detect small BALT nodules in non-engrafted NSG mice (Appendix A). However, large BALT structures composed mostly of human CD45^+^ lymphocytes were commonly present in the lung of NSG-BLT mice (Appendix A). Abundant HIV RNA were detected in hCD3^+^ T cells (Figure 1E) along with some hCD163^+^/CD68^+^ (Figure 1F) macrophages in those lymphoid aggregates. Thus, the lymphoid compartments of HIV-infected NSG-BLT mice contained many human CD3^+^ T cells expressing a high level of HIV RNA that were surrounded by numerous CD3^+^ T cells containing a single HIV RNA molecule (Figure 1A,C,E, inset). Occasionally, HIV-infected hCD68^+^CD163^+^ macrophages were detected with discrete HIV RNA signals in the nucleus and cytoplasm, which suggests active virus replication (Figure 1B,D, inset). GALT regions of rectosigmoid biopsies from untreated HIV-infected individuals [48] were used as a positive control for HIV detection (Figure 1G,H). Notably, rectosigmoid biopsies were highly enriched with CD68^+^CD163^+^ macrophages, and some of them contained a single fuchsia punctum in the nucleus (Figure 1H, inset) as did human macrophages in the lymphoid compartments of HIV-inoculated NSG-BLT mice (Figure 1B,D).

Taken together, these data indicate that HIV replicates robustly in the lymphoid compartments of NSG-BLT mice, similar to that in secondary lymphoid tissue in the human.

### 3.2. Fully Suppressive EFdA Treatment Impairs HIV RNA Production in the Lymphoid Tissues of NSG-BLT Mice

We reported that daily oral administration of the highly potent nucleoside analog EFdA at 10 mg/kg/day completely suppressed HIV viremia in NSG-BLT mice within 2 weeks of treatment [32]. These data prompted us to create an in vivo model of HIV persistence in which HIV replication in the lymphoid tissues of humanized mice is suppressed to extremely low levels, which recapitulated ART-suppression in HIV-infected individuals. The experimental approach of this model design is represented in Figure 2. One cohort of 30 NSG-BLT mice was generated using 8-week-old NSG mice followed by injection of autologous CD34^+^ cells 3 weeks later. The engraftment of mouse peripheral blood with human leukocytes was evaluated by flow cytometry 10 weeks after CD34+ cell injection (2 weeks after HIV_JR-CSF_ inoculation) (Figure 2A). Human CD45^+^ cells were found in mouse peripheral blood typically ranging from 15% to 30% of total mouse and human leukocytes with the proportions of CD4^+^ and CD8^+^ cells similar to those in human blood.

NSG-BLT mice were inoculated with HIV_JR-CSF_ 8 weeks after CD34+ cell injection, and, 2 weeks later, plasma and organs from 7 mice (group A) were collected to verify the establishment of HIV infection. HIV RNA levels in the plasma reached a mean of 3.4 log_10_ copies per 100 µL (Figure 3A,B, 2 weeks after inoculation), and 4 out of 7 spleens contained numerous HIV RNA^+^ cells with a mean of 1400 HIV RNA^+^ cells per mm^2^ (Figure 3C, 2 weeks after inoculation). Having validated well-established HIV infection in the mice 2 weeks after virus inoculation, the remainder of the HIV-inoculated mice were treated for 4 weeks with EFdA (10 mg/kg/day by once-daily oral gavage, groups B and C) or were untreated (groups E and F) beginning 2 weeks after HIV_JR-CSF_ inoculation (Figure 2B). Plasma and organs were collected 6 and 8 weeks after virus inoculation from HIV-inoculated EFdA-treated mice (groups B and C) and HIV-inoculated untreated mice (groups E and F).

Similar to our report [32], daily oral administration of EFdA at 10 mg/kg/day completely suppressed HIV viremia in NSG-BLT mice within 4 weeks of treatment, but HIV RNA levels in the plasma rebounded to the levels of untreated controls within 2 weeks after EFdA cessation (Figure 3A,B). We next used a combination of RNAscope ISH and IHC to investigate HIV-infected cells in the lymphoid compartments of NSG-BLT mice during this viral suppression and rebound. Human CD3^+^ and CD163^+^ cells containing HIV RNA were enumerated in 50 images of representative lymphoid white pulp regions of each spleen and expressed as the number of HIV RNA^+^ cells per mm^2^ of 5-µm tissue cross-section (Figure 3C). We found that spleens of completely suppressed mice contained a mean of 62 HIV RNA^+^ cells/mm^2^ 6 weeks after virus inoculation that was almost 25 times lower (*P* < 0.001) than that detected in the spleens of inoculated untreated mice (1500 cells/mm^2^). Although the majority of HIV RNA^+^ cells in the spleens of untreated mice were T cells containing a high level of HIV RNA, human CD163^+^CD68^+^ macrophages containing discrete fuchsia puncta in the nucleus (the sense strand of integrated HIV DNA) and cytoplasm (HIV RNA) were often detected (Figure 3D, inset). In contrast, spleens of EFdA-suppressed mice occasionally contained typically CD3^+^ T cells with a bright fuchsia dot in the nucleus, which indicates the detection of the sense strand of integrated HIV DNA (Figure 3E). Further analysis of NSG-BLT mouse spleens 2 weeks after EFdA cessation showed that, in accordance with HIV RNA levels in plasma (Figure 3B), the quantity of HIV RNA^+^ cells in the spleen dramatically increased in some, but not all, mice (Figure 3C). Notably, within 2 weeks after EFdA cessation, plasma from mouse #26 showed virus rebound to the level of the untreated controls (Figure 3B). The spleen from this mouse contained the greatest number of HIV RNA^+^ cells in the spleen at the same time point (Figure 3C,F).

Next, we compared the size of the cellular HIV reservoir during full EFdA suppression in the spleen (white pulp regions), lung (BALT regions), and Thy/Liv implant (Figure 4A,C,E,G). Since there is high variability in the density and total number of cells in different organs [49], we expressed the number of HIV RNA^+^ cells per 10^5^ cells [40] for this comparison instead of mm^2^, which was used in the evaluations of all samples that were from the same organ—the spleen. Although the average number of HIV RNA^+^ cells per 10^5^ cells was 2-fold less in lungs than in spleens and even less in Thy/Liv implants, there was no statistically significant difference between the groups (Figure 4A). In all three tissues, HIV RNA^+^ cells were rarely detected (Figure 4C,E,G). We found the same pattern of the highest number HIV RNA^+^ cells in the spleen and the lowest number in the Thy/Liv implant 2 weeks after EFdA cessation (Figure 4B). Two weeks after EFdA, the number of HIV RNA^+^ cells increased 30-fold in the spleens and lungs and 10-fold in the Thy/Liv implants, which demonstrates a dynamic viral spread in all evaluated lymphoid compartments. Notably, after EFdA cessation, we observed hCD163^+^ macrophages in all three organs with HIV signals in the nucleus, which signifies virus replication (Figure 4D,F,H). Based on these data, the spleens were chosen for further experiments.

Taken together, the results indicate that, despite fully suppressed HIV viremia in NSG-BLT mice by EFdA treatment, lymphoid tissues sustained a founder population of rare HIV RNA^+^ cells that enabled local virus expansion during the 2 weeks after drug cessation.

### 3.3. Fully Suppressive EFdA Treatment Does Not Eliminate Cells Harboring HIV DNA

The significant decline in the number of HIV RNA^+^ cells we observed in the spleen of EFdA-suppressed mice prompted us to examine the population of cells harboring HIV DNA. HIV DNA was detected by fluorescent (Figure 5A,C,D) and chromogenic (Figure 5B,E) DNAscope ISH using a target probe specific to the noncoding strand of integrated HIV DNA. HIV DNA was clearly distinguished as a discrete dot in the nucleus of numerous hCD3^+^ T lymphocytes (Figure 5A, yellow arrow) and random hCD163^+^ macrophages (Figure 5B, inset) in the spleen of HIV-inoculated untreated mice. Human T cells (Figure 5C, inset) and macrophages (Figure 5D,E, insets) containing an HIV DNA signal in the nucleus were also evident in the spleen of EFdA-suppressed mice. During treatment, there was no difference in the number of HIV DNA^+^ cells in spleens from untreated and EFdA-suppressed mice (Figure 5F). The same level of HIV DNA^+^ cells was maintained during the 2 weeks after drug cessation. These results reflect the feature of EFdA to inhibit HIV reverse transcriptase-catalyzed pro-viral DNA synthesis [50] and prevent new infection. Therefore, it suggests that the integration of the HIV genome into those HIV DNA^+^ cells occurred before EFdA treatment. Notably, the number of HIV DNA+ cells in the spleen of untreated mice continued to increase to almost 150 HIV DNA^+^ cells/mm^2^ 8 weeks after virus inoculation and was significantly higher than in the spleen of EFdA-suppressed mice after viral rebound (61 HIV DNA^+^ cells/mm^2^). Next, we verified that cells harboring HIV DNA during EFdA suppression contain a replication competent virus. Immediately after euthanasia, each mouse spleen was cut in half with one half fixed for RNAscope assays and the other half was used for isolation of spleen cells. When dispersed spleen cells were co-cultivated with PHA-stimulated normal donor PBMC for 7 days, similar levels of infectious HIV (3.0–3.5 log_10_ IUPM) were detected in the untreated and EFdA-suppressed groups (Figure 5G) even though we assume that greater IUPM should be present in the spleens of untreated mice because the upper limit of IUPM detection was 3.75 log_10_.

Thus, despite potent EFdA suppression at the transcriptional level, spleen cells from HIV-infected NSG-BLT mice maintained consistent levels of integrated HIV DNA and are capable of producing a replication-competent virus.

### 3.4. A Rare Subset of HIV-Infected Cells Expressed HIV RNA and p24 Capsid Protein during Fully Suppressive EFdA Treatment

To further characterize the cellular reservoir of HIV during EFdA suppression, we combined the detection of HIV RNA by RNAscope ISH and p24 capsid protein by IHC. In this experiment, HIV RNA^+^ cells and p24^+^ cells were enumerated in 30 images of randomly selected representative lymphoid white pulp regions of each spleen and expressed as a number of HIV RNA^+^ cells per mm^2^ of 5-µm tissue cross-section (Figure 6A) or as a percentage of HIV p24^+^ cells relatively to the total number of HIV RNA^+^ cells (Figure 6B). Similar to the experiment described above (Figure 3C), we detected a small population of cells containing HIV RNA (a mean of 29 cells/mm^2^) in the spleens of EFdA-suppressed mice 6 weeks after HIV inoculation (Figure 6A). Notably, about 15% of those cells expressed the p24 protein (Figure 6B,C). This number declined to 1% when the outlier (mouse #1) was excluded. However, in some spleens (mouse #3, #14, #44), almost 50% of the HIV RNA^+^ cells were p24^+^ 2 weeks after EFdA cessation (Figure 6B,D). In those spleen, we identified local expansion of an HIV^+^ founder population that is outlined by a dotted white line in the inset (Figure 7A,B). The yellow color in the merged inset indicates spatial overlap between the red HIV RNA signal and the green p24 signal. It represents an HIV^+^ founder cell (Figure 7B). Numerous red puncta outlined by the dotted white line are single HIV RNA molecules that may represent spreading putative HIV virions.

In spleens from the untreated NSG-BLT mice, the majority of cells expressing a high level of HIV RNA was identified as human CD3^+^ T cells (Figure 7C). These HIV RNA^high^ cells were often surrounded by human CD163^+^ macrophages containing discrete HIV RNA^+^ red dots in the nucleus and cytoplasm. Almost all the HIV RNA^+^ cells in the spleens of EFdA-suppressed mice were also identified as CD3^+^ T cells (Figure 3E and Figure 4C). We performed an extensive co-localization analysis for HIV RNA, p24, and human CD163 or CD3 with 6 spleen cross-sections from each of 6 EFdA-suppressed mice in order to detect very rare hCD163^+^ macrophages expressing both HIV RNA and p24 (Figure 7D). However, the level of expression of both HIV RNA and p24 was less than was observed in macrophages 2 weeks after EFdA cessation (Figure 7E).

Altogether, these results revealed evidence for a transcriptionally and translationally competent HIV reservoir in the lymphoid compartments of EFdA-suppressed mice.

### 3.5. During Fully Suppressive EFdA Treatment, HIV RNA and Integrated Provirus Were Detected in CD163^+^ Macrophages

Having detected very unique HIV RNA^+^p24^+^ macrophages in the mouse organs, we next performed a comprehensive analysis of HIV RNA expression specifically in macrophages (Figure 8A). HIV RNA^+^ cells with and without hCD163^+^ expression were enumerated, and the number of HIV RNA+ macrophages was expressed as a percentage of total HIV RNA^+^ cells per image acquired at a magnification of 630×. Although the percentage of macrophages containing HIV RNA in the spleens of EFdA-suppressed mice declined with EFdA treatment, there was no statistical difference between the groups with HIV RNA^+^ macrophages representing means of 3% to 9% of total HIV RNA^+^ cells. The number of human CD163^+^ macrophages containing integrated HIV DNA was slightly higher, and the means varied from 6% to 14% of total HIV DNA^+^ cells (Figure 8B). No statistical differences between groups were found by the Mann–Whitney U-test.

Two types of HIV^+^ macrophages were frequently observed in the spleens of untreated mice. First, hCD163^+^ macrophages contained both distinct fuchsia dots in the nucleus by visualizing the sense strand of integrated HIV DNA and solid red dots of HIV RNA in the cytoplasm, which indicates HIV replication (Figure 8C). The second type of HIV^+^ macrophage contained only a strong red signal of integrated HIV DNA in the nucleus with no HIV RNA detected in the cytoplasm (Figure 8D). This second type of macrophage was commonly detected in the spleens during EFdA suppression. The DNAscope further verified the signal of integrated provirus observed in the nucleus of macrophages by the RNAscope. We incorporated an RNase pretreatment step into the DNAscope assay to ablate HIV RNA detection and used an HIV-1 clade B sense probe that specifically targets integrated viral DNA noncoding strands [40]. The yellow arrows show human macrophages with a bright red dot of HIV DNA located in the blue nucleus stained with hematoxylin (Figure 8E,F). The ability to detect the sense strand of integrated HIV DNA using an HIV *gag-pol* antisense probe allowed us to detect potential HIV spread from a CD3^+^ T cell expressing a high level of HIV RNA to an adjacent hCD163^+^ macrophage with a single pink dot in the nucleus resulting from the spatial overlap of the red HIV DNA signal with the blue DAPI nucleus counterstain (Figure 9A). These data demonstrated that CD163^+^ macrophages in HIV-inoculated NSG-BLT mice may become infected by HIV and prompted us to examine further viral spread after EFdA cessation. The innate immune receptor DC-SIGN (dendritic cell-specific intercellular adhesion molecule-3 grabbing non-integrin) is a well-known enhancer of HIV infection that binds to the HIV envelope glycoprotein gp120 and promotes efficient infection in *trans* of cells that express CD4 and CCR5 [51]. The DC-SIGN-dependent mechanism of macrophage-HIV virion engagement is essential for both macrophage *cis* and *trans* infection [52]. We found that human CD163^+^ macrophages, which were frequently detected in proximity to HIV-infected T cells, express DC-SIGN protein (Figure 9B, yellow arrow). These data were observed in 5 spleens in a total of 3 multiplex fluorescent RNAscope/IHC assays. Next, we performed co-localization analysis for HIV RNA, human DC-SIGN, and human CD3 in 6 multiplex fluorescent RNAscope/IHC assays using 5 spleens from EFdA-suppressed mice 2 weeks after EFdA cessation. Detailed tissue analyses allowed us to monitor the following steps of HIV spread in vivo after EFdA cessation and viral rebound: (1) HIV RNA accumulation along the CD3^+^ cell membrane of T cells was often accompanied by DC-SIGN clustering on the cell surface of adjacent presumptive macrophages, which resembles DC-SIGN expression at the virological synapse [53,54,55] (Figure 9C, yellow arrows). (2) At the site of surface contact between an hCD3^+^ T cell (white) and a DC-SIGN^+^ presumptive macrophage, HIV virions occasionally appeared with a yellow color that results from the spatial overlap of a red HIV RNA punctum and green DC-SIGN protein, which resembles virion transport across the virological synapse [56,57] (Figure 9D). (3) DC-SIGN was highly expressed at the site of surface contact between an HIV-infected T cell abundantly expressing viral RNA and adjacent macrophages with evidence of initial HIV infection. This suggests an HIV spread (Figure 9E). (4) Abundant HIV RNA expression by infected DC-SIGN^+^CD3^–^ presumptive macrophages. Figure 9F shows discrete HIV RNA puncta likely representing HIV virions that are redistributed from the HIV RNA^high^ cell body (yellow arrow) to the DC-SIGN^+^ cell periphery (white arrow).

Together, these results demonstrate that the spleen of HIV-inoculated NSG-BLT mice is enriched with CD163^+^ macrophages and that these macrophages are capable of harboring HIV during EFdA suppression. Furthermore, during the 2 weeks after EFdA cessation, they form numerous surface site contacts with T cells expressing a DC-SIGN that seem to be involved in the viral spread.

## 4. Discussion

In this study, we developed an in vivo model of HIV persistence in which viral replication in the lymphoid compartments of humanized mice was inhibited by the highly potent HIV reverse transcriptase inhibitor EFdA to the level of single cells recapitulating ART-suppression in HIV-infected individuals. Investigating the source, composition, and size of the persistent HIV reservoir provides important information about cellular subsets that need to be targeted to facilitate HIV eradication. Tracking the earliest steps of HIV spread from these cellular reservoirs after drug cessation is pivotal for elucidating how infection can be prevented. Therefore, our goal was to identify HIV-infected cells at the single-cell level and to investigate their spatial relationship with neighboring cells within lymphoid tissues during active infection, suppression with EFdA, and after drug cessation. Expanding on our previous report [32], a humanized mouse model of HIV persistence was generated using NSG-BLT mice. For more than a decade, humanized BLT mice have been successfully employed for the study of HIV pathogenesis, latency, and preclinical evaluation of multiple antiretroviral therapies [30,58,59,60,61,62,63,64,65,66,67,68,69]. In agreement with all these studies, we show in this study that NSG-BLT mice display well-differentiated lymphoid compartments of human origin and are highly susceptible to HIV infection (Figure 1). HIV infection in NSG-BLT mice recapitulated key features of HIV infection in the human, including high-titer viremia in untreated mice, suppression of viremia when treated with a highly potent inhibitor of HIV replication, and viral rebound after a treatment interruption.

A significant reduction of cell-associated HIV RNA in lymphoid tissues including bone marrow, Thy/Liv implant, lymph node, liver, lung, and spleen was achieved when BLT mice were treated with EFdA orally administrated at 10 mg/kg/day [68]. However, despite comprehensive quantitative evaluation of HIV RNA levels in purified human cell populations by real-time RT-PCR, the approach used in this study lacked crucial information about viral reservoirs within their native tissue microenvironments comprising distinct cellular compositions and functions. Therefore, our goal was to investigate the localization and phenotypic features of HIV reservoir cells within their tissue milieu and to track viral spread in vivo after drug cessation. We employed a combination of IHC and state-of-the-art RNAscope ISH with single-molecule resolution to focus our study on the human lymphoid compartments developed in NSG-BLT mice for the following reasons. First, despite the fact that peripheral blood reflects events in lymphoid tissues, T cells in the circulating blood represent only about 2% of the total body T-cell pool, and the largest portion of lymphocytes are in the human spleen and lymph nodes with 1% to 9% of total lymphocytes located in the gut lamina propria [70]. Second, in patients on ART, HIV RNA is detected at higher levels in primary lymphoid tissue (tonsil and bone marrow) [42] and secondary lymphoid tissues such as GALT [43,44] and lymph nodes [45]. In addition, the human lung contains a similar number of CD4^+^ and CD8^+^ lymphocytes as the gut [70]. In our study, numerous lymphoid compartments were detected in the lung of NSG-BLT mice that were composed entirely of human cells (Appendix A). An accumulation of B and T lymphocytes intermingled with macrophages around high endothelial venules in the wall of the bronchi is called bronchus-associated lymphoid tissue (BALT) [71]. BALT was first described in the lung of rabbits more than 30 years ago [72]. Furthermore, it represents a normal component of the lung’s immune system in many animals. The prevalence of BALT differs between species, but it is largely absent in healthy humans and mice [47,73] and can be classified as a tertiary lymphoid organ [71]. We found that, in NSG-BLT mice, BALT represented a key lymphoid structure for study of the HIV reservoir. The distribution of HIV RNA^+^ cells in the BALT regions of untreated NSG-BLT mice (Figure 1E,F) was remarkably similar to that in the GALT regions of rectosigmoid biopsies from untreated HIV-infected individuals (Figure 1G,H). During EFdA suppression, the frequency of very rare HIV RNA^+^ cells in the BALT region of the lung was comparable to those observed in the lymphoid white pulp of the spleen and Thy/Liv implant (Figure 4). HIV spread during the 2 weeks after EFdA cessation was also consistent in the human lymphoid compartments of the mouse spleen, lung, and Thy/Liv implant, which is indicated by comparable numbers of HIV RNA^+^ cells/mm^2^. These data reflect the similar cellular compositions of the humanized lymphoid compartments in different NSG-BLT mouse organs, which contain many CD4^+^ T cells positioned in close proximity to each other facilitating cell-to-cell spread of HIV [74,75]. In chronic infection, HIV replicates mainly in the lymphoid tissue [5,11,76], and splenic T cells contain an average of 3.2 integrated proviruses per cell [77], which indicates cell-associated transmission of a high multiplicity of infection. In accordance with these reports, a small transcription-competent HIV reservoir was clearly established in the spleen of EFdA-suppressed mice (Figure 3C,E, and Figure 4C). Furthermore, fully suppressive EFdA treatment for 4 weeks did not eliminate cells harboring HIV provirus in the spleen (Figure 5F), and quantitative viral outgrowth assays with dispersed spleen cells indicated the presence of replication-competent HIV (Figure 5G).

These results are consistent with a recently reported examination of HIV reservoirs in lymph node and gut biopsies after at least 2 years of ART and SIV reservoirs in necropsy tissues from ART-suppressed rhesus macaques [78]. RNAscope ISH technology and quantitative image analysis were used to define HIV and SIV tissue reservoirs on the basis of the number vRNA^+^, vDNA^+^, and virus-producing cells. Similar to our data on HIV reservoirs in the lymphoid compartments of humanized mice, Estes et al. detected a prominent consistency in the frequency of vRNA^+^ and vDNA^+^ cells among lymph nodes from different anatomical locations. Moreover, the highest number of cells expressing SIV RNA was detected in lymphoid tissues (lymph nodes, spleen, and GALT) and the lung. ART administered for 20–22 weeks decreased the frequency of SIV RNA^+^ cells by approximately 2 log_10_ in lymph nodes but only about two-fold in the gut and spleen. In NSG-BLT mice, EFdA-treatment for only 4 weeks decreased the number of HIV RNA^+^ cells in the spleen by 25-fold, which underscores the great effectiveness of this novel and exceptionally long-lasting and potent HIV drug currently being developed by Merck as MK-8591. On the other hand, EFdA treatment did not significantly impact the quantity of HIV DNA^+^ cells (Figure 5F) and the infectious virus (Figure 5G). In the rhesus macaque study, ART decreased the mean frequency of vDNA^+^ cells in the lymphoid tissues by 1.7 log_10_, and the frequency of HIV DNA^+^ cells declined by about 3 log_10_ in individuals who had been ART treated for >2 years. This may suggest that the elimination of vDNA^+^ cells in HIV-inoculated NSG-BLT mice will require further optimization of the EFdA treatment regimen or that various lymphoid tissues retain different drug concentrations [3,5]. No changes in the size of the HIV DNA^+^ cell reservoir were observed in gut tissues from ART-suppressed rhesus macaques [78].

Thus, our results are consistent with the generally accepted concept that suppressive ART does not completely suppress HIV production in tissues. We report in this paper that, during highly potent EFdA therapy, lymphoid tissues of NSG-BLT mice harbor a rare subset of cells producing both HIV RNA and low levels of p24 protein (Figure 6A–C and Figure 7D), which represents the translation-competent reservoir [79]. These cells spontaneously produce viral products in the absence of a spreading infection and may rapidly reseed HIV infection after drug cessation, which we have observed as a 50-fold increase in the frequency of HIV RNA^+^p24^+^ cells during the 2 weeks after EFdA cessation (Figure 6B,D and Figure 7A,B). Despite the fact that not all translation-competent proviruses are also replication competent [80,81], it was suggested that the translation-competent cellular reservoir must be considerably enriched to be replication competent [82]. Consequently, identification of translation-competent cellular reservoirs was performed by the simultaneous detection of the HIV protein and HIV RNA by an RNAflow technique in samples from untreated and ART-treated individuals after in vitro re-stimulation [79,83]. We further expanded this approach by adding the spatial localization of HIV reservoir cells within their tissue milieu. In spleens two weeks after EFdA cessation, we detected a distinctive pattern of a viral spread represented by red puncta of HIV RNA^+^ putative virions releasing from a cell that was HIV RNA^+^ and p24^+^. These data are consistent with the suggestion [82] that single cells containing transcription/translation-competent viruses and producing viral proteins during ART may represent the cell population from which an HIV rebound may occur.

The work presented in this paper also focused on macrophages as an intriguing source of an infectious virus during suppressed HIV infection and on interaction of macrophages with T cells during drug cessation. Although HIV infection of macrophages was reported more than three decades ago [84], the study of HIV cellular reservoirs that support viral persistence focused mainly on CD4^+^ T cells, and the role of macrophages in HIV persistence continues to be controversial. We and others have demonstrated the presence of HIV RNA^+^ or HIV DNA^+^ macrophages in the gut of ART-suppressed HIV-infected individuals [48,85,86]. HIV infection of macrophages has been studied intensively in humanized mouse models [30,87]. The ability of HIV to infect and persist in macrophages during ART was recently demonstrated in humanized myeloid-only mice [88]. In accordance with these observations, we report that lymphoid tissues of HIV-inoculated NSG-BLT mice are enriched with human CD163^+^ macrophages that are able to harbor HIV during EFdA suppression, which is indicated by nuclear localization of the integrated provirus (Figure 5D,E and Figure 8E,F). Moreover, we show in this study for the first time that some of those human CD163^+^HIV RNA^+^ macrophages are able to maintain a low level of p24 production during ART suppression (Figure 7D). This finding suggests that very rare CD163^+^ macrophages containing transcription/translation-competent viruses and also producing p24 may promote HIV recurrence if ART is stopped. Despite the fact that the frequency of both HIV RNA^+^ and HIV DNA^+^ macrophages was not changed during the two weeks after the drug cessation (Figure 8A,B), these cells increased in p24 production (Figure 7E). SIV infection promoted a significant increase in CD163^+^ macrophages in the spleen of pigtailed macaques, and they were highly susceptible to SIV [89].

Macrophages participate in HIV dissemination by various mechanisms of cell-to-cell transfer including selective capture and engulfment of HIV-infected T cells leading to efficient infection of the macrophage [90], heterotypic envelope-dependent cell fusion leading to the formation of lymphocyte-macrophage fused cells [91], and virological synapse formation [74,75,92,93]. Virological synapses are transient, polarized zones of surface contact between HIV-infected and uninfected cells where viral particles are concentrated and transferred from cell-to-cell [54]. Notably, in our study, CD163^+^ macrophages expressed DC-SIGN at the surface contact points with presumptive T cells expressing a high level of HIV RNA (Figure 9B). Numerous sites of DC-SIGN^+^ cell contact formed by CD3^+^DC-SIGN^–^ T cells and CD3^–^DC-SIGN^+^ presumptive macrophages were observed (Figure 9C–F). DC-SIGN, which is a mannose-specific C-type lectin receptor with a high affinity for the intercellular adhesion molecule ICAM, is expressed on the surface of macrophages, dendritic cells, and B cells [53]. The DC-SIGN expressed on the surface of dendritic cells efficiently captures HIV virions in the periphery by binding to gp120 and transporting them to target T cells in the secondary lymphoid organs. It enhances their *trans* infection by forming virological synapses [51,94]. Virological or infectious synapses are adhesive structures formed between infected and uninfected cells that enhance HIV cell-to-cell spread [74,75,92,93]. After encountering target CD4^+^ T cells, virions in dendritic cells re-localize to sites of DC-SIGN^+^ cell contact at the virological synapse and efficiently transfer the virus [57]. The latter is very intriguing because, during the two weeks after EFdA cessation, we observed numerous sites of DC-SIGN^+^ cell contact co-localized with concentrated HIV RNA puncta (Figure 9C–E). These in vivo data reinforce the role of the DC-SIGN in HIV pathogenesis [51] and highlight a mechanism that could be crucial for developing strategies to prevent or block further HIV infection. It is of interest that the macrophage-mediated HIV *trans* infection of autologous CD4^+^ T cells is associated with the control of disease progression. It is impaired in HIV-infected non-progressors [52]. Furthermore, the inability of macrophages to *trans* infect autologous T cells in non-progressors was found to be cholesterol-dependent and related to a low expression of the DC-SIGN.

In summary, we have established in this study a humanized mouse model of HIV persistence using NSG-BLT mice and a highly potent HIV reverse transcriptase inhibitor EFdA with the goal of recapitulating key features of HIV persistence in humans. We used a combination of state-of-the-art ISH technology (RNAscope) and immunohistochemistry to quantitatively investigate the composition and distribution of HIV reservoirs within lymphoid tissues. We found that the lymphoid compartments of EFdA-suppressed humanized mice harbor very rare transcription/translation-competent HIV reservoirs that may enable viral rebound. Although CD3^+^ T cells constituted a large component of the HIV reservoir, a small subset of CD163^+^ macrophages harboring HIV RNA or DNA was also identified. Our finding that very rare HIV RNA^+^ macrophages are capable of producing low levels of p24 in vivo during EFdA suppression indicates that CD163^+^ macrophages might support HIV persistence during ART suppression. The monitoring of the HIV spread in the NSG-BLT mouse spleen two weeks after EFdA cessation further suggests that highly expressed DC-SIGN on macrophages at the site of cell contact co-localizing with concentrated HIV RNA puncta could be involved in the formation of virological synapses, which facilitates efficient viral spread and rapid viral rebound. Targeting the ability of macrophages to mediate highly efficient HIV *trans* infection of CD4^+^ T cells could be a promising therapeutic approach for eliminating the viral reservoir.

## Figures and Tables

**Figure 1 viruses-11-00256-f001:**
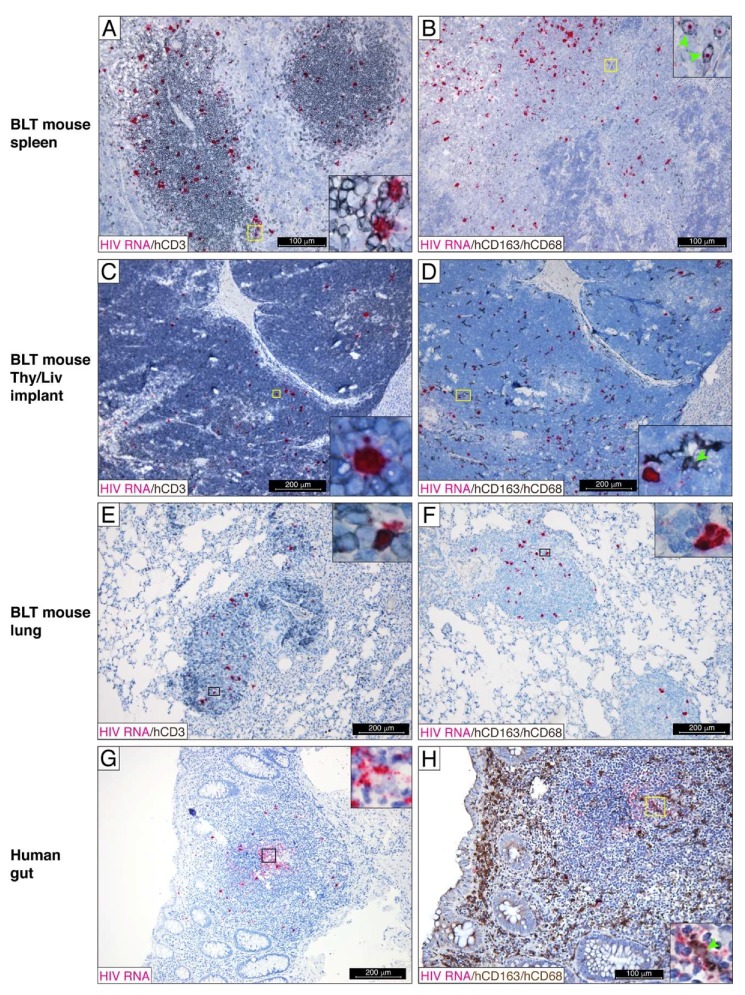
Tissue distribution of human HIV RNA^+^ cells in the lymphoid compartments of HIV-viremic NSG-BLT mice. Humanized NSG-BLT mice were inoculated intraperitoneally with 25,000 TCID_50_ HIV_JR-CSF_ and euthanized 2 weeks later for tissue collection. HIV RNA was detected by RNAscope ISH using an HIV-*gag-pol* antisense probe in the mouse spleen (**A**,**B**), Thy/Liv implant (**C**,**D**), and mouse lung (**E**,**F**). Rectosigmoid biopsies from an untreated HIV-infected individual were used as a positive control for HIV detection (**G**,**H**). The RNAscope assay was followed by chromogenic immunohistochemistry for the human T-cell marker hCD3 (**A**,**C**,**E**) and the human macrophage markers hCD68 and hCD163 (**B**,**D**,**F**,**H**). Nuclei were counterstained with hematoxylin. Yellow (**A**–**D**,**H**) and black (**E**–**G**) oblongs indicate regions magnified in insets. The green arrowhead shows macrophages containing a single fuchsia punctum in the nucleus. The results in (**A**–**F**) are representative of those observed in the spleen, Thy/Liv implant, and lungs from 6 untreated HIV-inoculated NSG-BLT mice. The results in (**G**) and (**H**) are representative of those observed in rectosigmoid biopsies from 3 untreated HIV-infected individuals.

**Figure 2 viruses-11-00256-f002:**
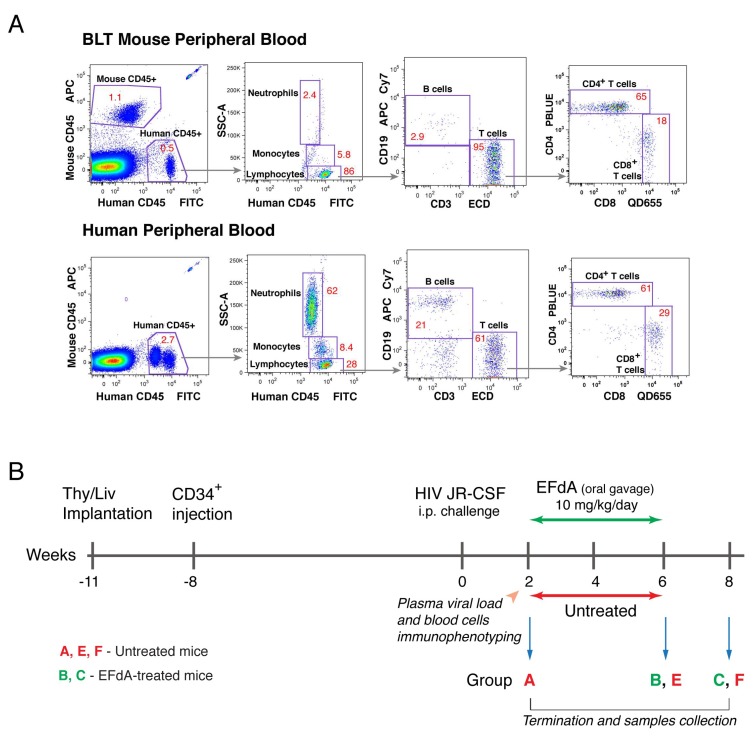
Experimental approach to create an in vivo mouse model of HIV suppression recapitulating ART-suppression in HIV-infected individuals. (**A**) Representative engraftment of NSG-BLT mouse peripheral blood with human leukocytes. Trucount enumeration by flow cytometry was used to determine the absolute number of human CD3^+^, CD4^+^, CD8^+^ T cells, B cells, monocytes, and neutrophils per μL of NSG-BLT mouse (upper diagram) and human (low diagram) peripheral blood. (**B**) Schematic diagram of the experimental approach. At the time of intraperitoneal HIV_JR-CSF_ inoculation (11 weeks after Thy/Liv implantation/8 weeks after CD34^+^ cells injection), NSG-BLT mice (30 from a single donor) had appropriate levels of human leukocyte reconstitution in the peripheral blood, which is shown in panel (**A**). Two weeks after HIV_JR-CSF_ inoculation, plasma and organs were collected from 7 mice (group A) to verify the establishment of HIV infection. The remaining HIV-inoculated mice were treated for 4 weeks with EFdA (10 mg/kg/day by once-daily oral gavage) or were untreated beginning 2 weeks after HIV_JR-CSF_ inoculation. Plasma and organs were collected 6 and 8 weeks after virus inoculation from HIV-inoculated EFdA-treated mice (groups B and C) and HIV-inoculated untreated mice (groups E and F). Plasma and organs from uninoculated untreated mice were used as negative controls.

**Figure 3 viruses-11-00256-f003:**
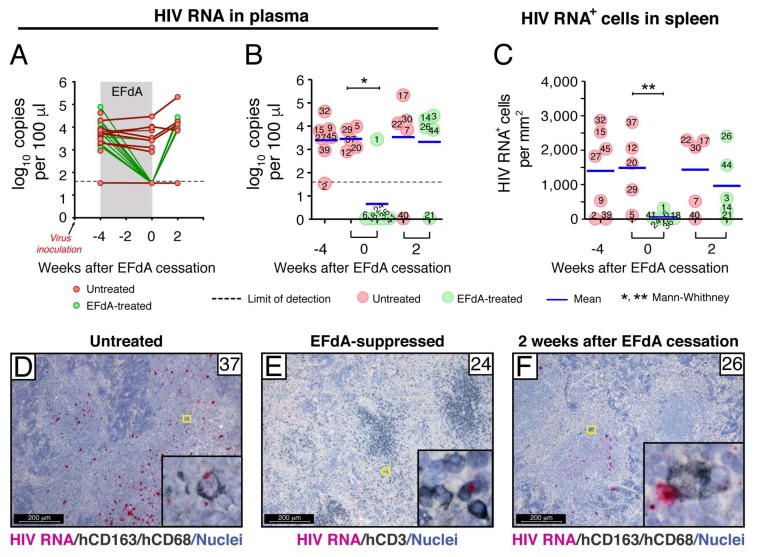
Effect of EFdA administration on the level of HIV RNA^+^ cells in NSG-BLT mouse spleens. HIV RNA levels in NSG-BLT mouse plasma by the Abbott RealTime HIV-1 Viral Load (PCR) assay represented by a time-course in all individual mice (**A**) or by the actual value for each individual mouse per group (**B**). RNAscope ISH using an HIV-gag-pol antisense probe detected HIV RNA+ cells in NSG-BLT mouse spleens (**C**). Fifty images of representative white pulp regions containing human CD3^+^ and CD163^+^ cells were acquired at a magnification of 630× for each spleen. HIV RNA^+^ cells were enumerated by ImageJ software. The results of 3 RNAscope assays with 3 tissue sections per spleen are represented as the mean value in panel (**C**). The Mann–Whitney U-test performed a statistical comparison between groups and was considered significant at * *P* < 0.05 (**B**) and ** *P* < 0.01 (**C**). Phenotyping of HIV RNA^+^ cells was accomplished by combined RNAscope in situ hybridization and chromogenic IHC for the human macrophage markers hCD163 and hCD68 (**D**,**F**) and human T-cell hCD3 (**E**). HIV RNA in the panel (**D**) and (**E**) insets was visualized by alkaline phosphatase-Fast Red fluorescence detection. Nuclei were counterstained with hematoxylin. Yellow (**D**–**F**) rectangles outlined the region of interest and was magnified in the insets. Representative images of spleens from untreated mouse #37 (**D**), EFdA-suppressed mouse #24 (**E**), and mouse #26 2 weeks after EFdA cessation (**F**).

**Figure 4 viruses-11-00256-f004:**
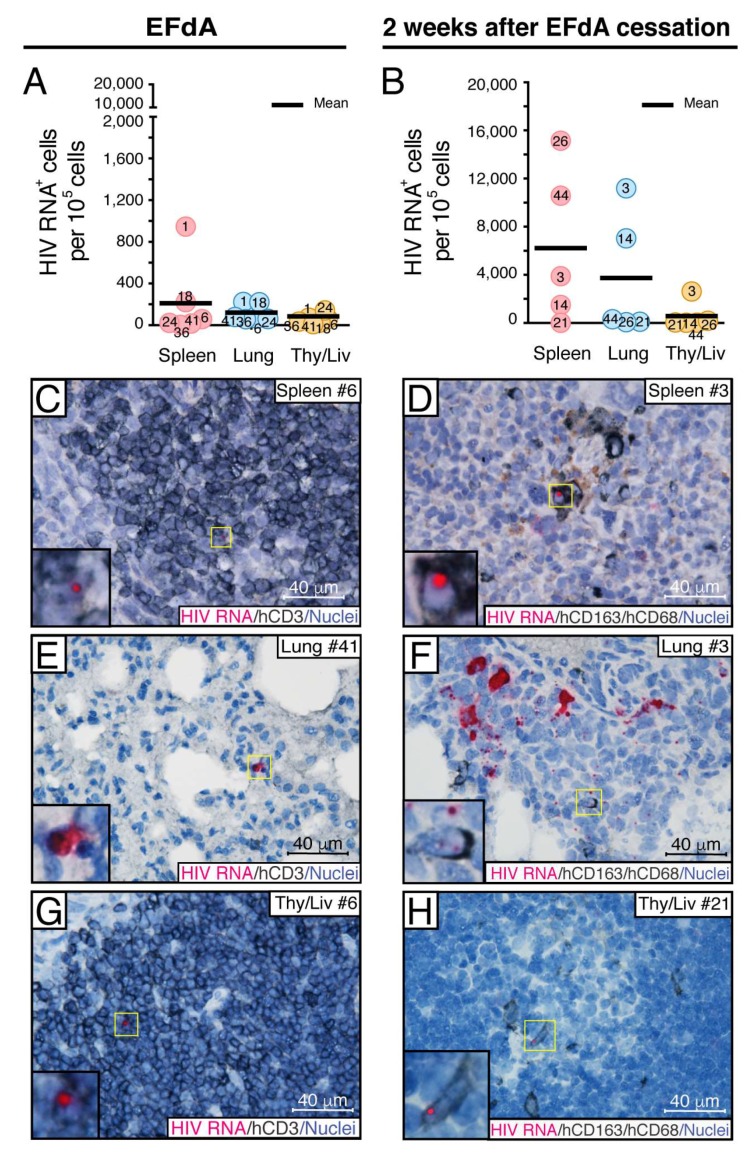
Identification of HIV RNA^+^ cellular compartments in the spleen, lung, and Thy/Liv implant of NSG-BLT mice during fully suppressive EFdA treatment and after drug cessation. HIV RNA^+^ cells were detected by RNAscope ISH using a HIV-*gag-pol* antisense probe in NSG-BLT mouse spleens, lungs, and Thy/Liv implants at the time of EFdA cessation (**A**,**C**,**E**,**G**) and 2 weeks later (**B**,**D**,**F**,**H**). The results are representative of 3 RNAscope assays with 3 tissue sections per organ. (**A**,**B**) Fifty images of representative region containing human lymphoid and myeloid cells were acquired at a magnification of 630× for each organ. HIV RNA^+^ cells were enumerated by ImageJ software and expressed per 10^5^ cells. The mean value for each organ is represented in panels (**A**) and (**B**). The Mann−Whitney U-test performed the statistical comparison between organs at the same time point and was not significant (*P* > 0.05). Phenotyping of HIV RNA^+^ cells was accomplished by combined RNAscope ISH and chromogenic immunohistochemistry for human T-cell hCD3 (**C**,**E**,**G**) and human macrophage markers hCD163 and hCD68 (**D**,**F**,**H**). Panels (**C**,**E**), and (**G**) show representative images of organs from EFdA-suppressed mice, and panels (**D**,**F**), and (**H**) show representative images of organs from mice 2 weeks after EFdA cessation. Organ numbers in (**C**–**H**) indicate mouse numbers in panels (**A**) and (**B**). HIV RNA in the insets in panels (**C**,**D**,**G**), and (**H**) was visualized by alkaline phosphatase-Fast Red fluorescence detection. Nuclei were counterstained with hematoxylin. Yellow (**C**–**H**) rectangles show regions magnified in the insets.

**Figure 5 viruses-11-00256-f005:**
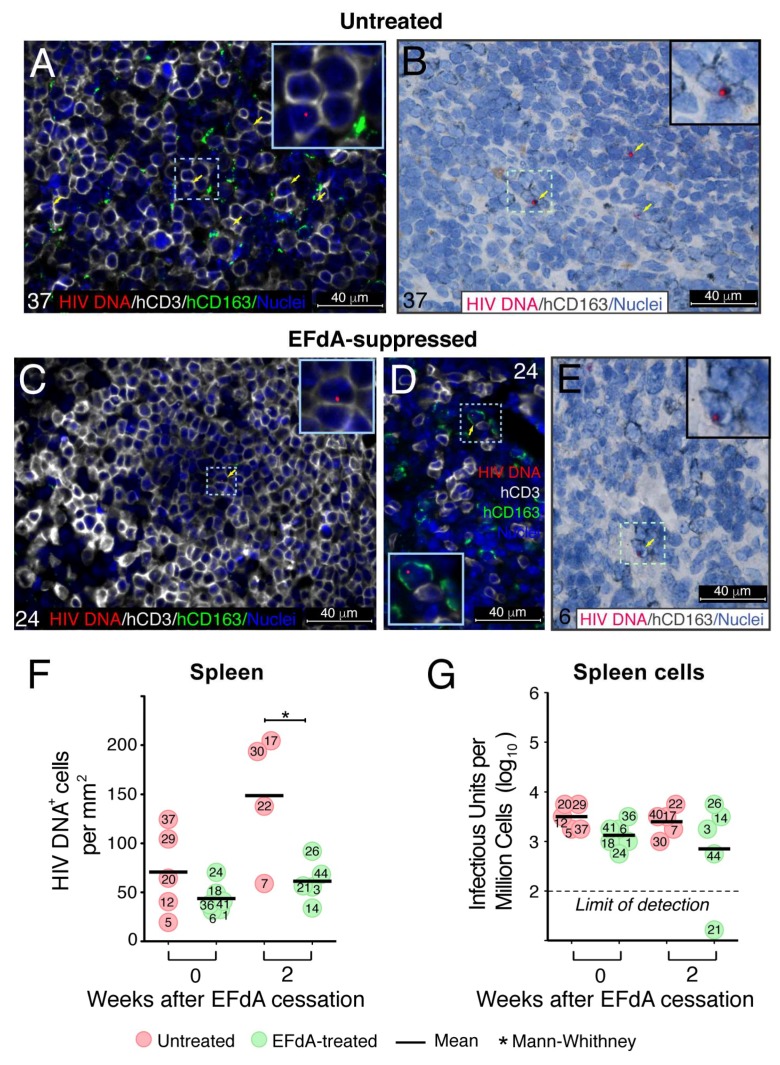
Fully suppressive EFdA therapy does not eliminate HIV DNA^+^ cells in the spleen. HIV DNA was detected in NSG-BLT mouse spleens by a DNAscope multiplex fluorescent (**A**,**C**,**D**) and chromogenic (**B**,**E**,**F**) assays using an HIV-1 clade B sense probe. DNAscope labeling of HIV DNA was combined with fluorescent co-immunolabeling for human CD3 (hCD3) and CD163 (hCD163) (**A**,**C**,**D**) or chromogenic immunolabeling for CD163 (hCD163) (**B**,**E**). Nuclei were counterstained with DAPI (**A**,**C**,**D**) or hematoxylin (**B**,**E**). HIV DNA in panels (**B**) and (**E**) was visualized by alkaline phosphatase-Fast Red fluorescence detection. The region of interest is outlined by dashed light blue rectangles (**A**–**E**) and magnified in the insets. The results are representative of those observed in 5 or 6 spleen samples in untreated (**A**,**B**) and EFdA-suppressed groups (**C**,**D**,**E**). (**F**) Fifty images of the representative white pulp regions containing human CD3^+^ and CD163^+^ cells were acquired at a magnification of 630× for each spleen. HIV DNA^+^ cells were enumerated by ImageJ software. The results of 3 DNAscope assays with 3 tissue sections per spleen are represented as the mean value. (**G**) Infectious units per million (IUPM) cells assay with dispersed spleen cells. The IUPM assay was performed with 32,000, 10,000, and 3200 spleen cells co-cultivated with phytohemagglutinin-stimulated PBMC for 7 days, and IUPM were determined by detection of p24 in supernatants. The upper limit of detection is, therefore, 3.75 log_10_ infectious units per million spleen cells and is likely greater for 2 of the untreated mice. The Mann−Whitney U-test performed a statistical comparison between the groups and was considered significant at * *P* < 0.05 (**F**). Organ numbers 37 (**A**,**B**), 24 (**C**,**D**), 6 (**E**) indicate mouse numbers shown in panels (**F**) and (**G**).

**Figure 6 viruses-11-00256-f006:**
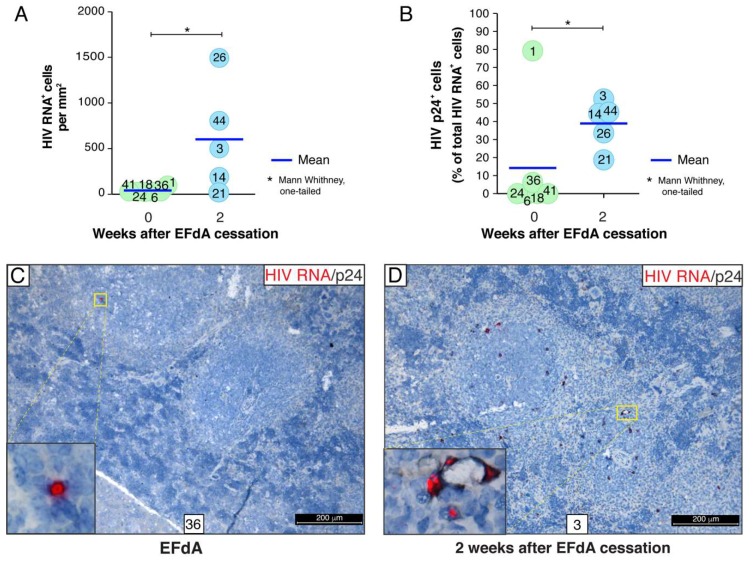
Translation-competent HIV reservoir was occasionally detected in the NSG-BLT mouse spleen during fully suppressive EFdA treatment. (**A**) HIV RNA^+^ cells were detected by RNAscope ISH using an HIV-*gag-pol* antisense probe in NSG-BLT mouse spleens at the time of EFdA cessation (0) and 2 weeks later. RNAscope ISH was followed by chromogenic IHC for HIV p24 (**B**). Thirty images of representative white pulp regions containing human lymphoid and myeloid cells were acquired at a magnification of 630× for each spleen. HIV RNA^+^ and p24^+^ cells were enumerated by ImageJ software. The number of HIV p24^+^ cells was normalized to the number of HIV RNA^+^ cells per image and expressed as a percent. A statistical comparison between groups was performed by the Mann−Whitney U-test and was considered significant at * *P* < 0.05 with the one-tailed test. (**C**,**D**) Combined RNAscope ISH for HIV RNA and chromogenic IHC for HIV p24. HIV RNA in the insets was visualized by alkaline phosphatase-Fast Red fluorescence detection. Organ numbers in (**C**) and (**D**) indicate mouse numbers shown in panels (**A**) and (**B**). Nuclei were counterstained with hematoxylin. Yellow rectangles indicate regions magnified in the insets.

**Figure 7 viruses-11-00256-f007:**
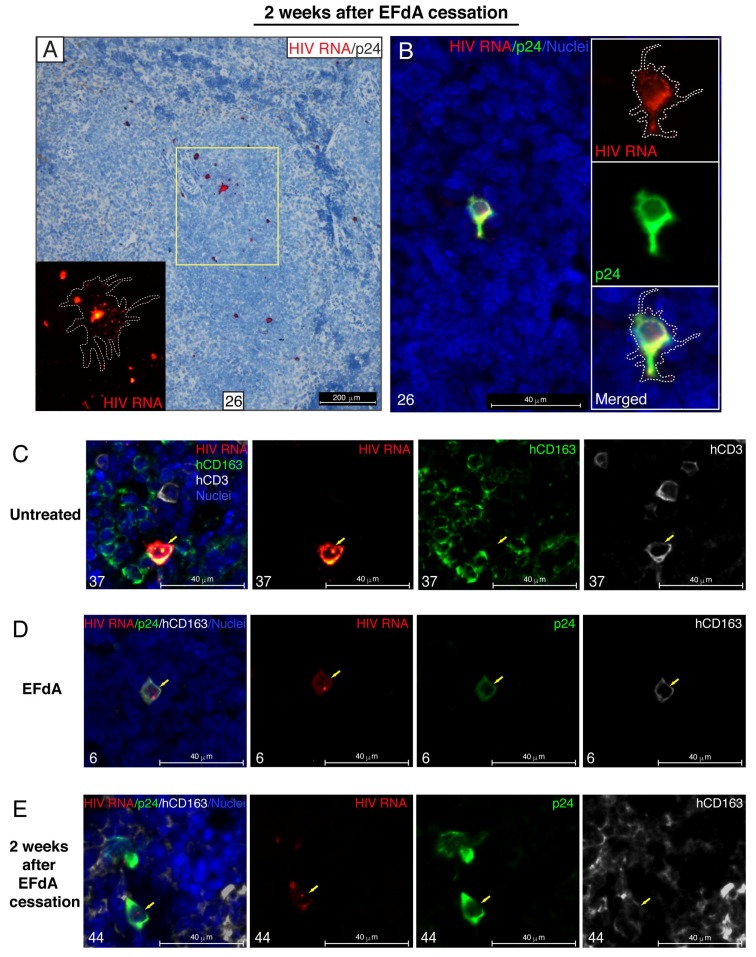
HIV RNA^+^ macrophages can maintain a low level of HIV p24 expression during EFdA suppression. (**A**,**B**) HIV appears to spread efficiently from infectious foci in the spleen 2 weeks after EFdA cessation. (**A**) Combined RNAscope ISH for HIV RNA and chromogenic IHC for HIV p24. HIV RNA was visualized by alkaline phosphatase-Fast Red fluorescence detection. The yellow rectangles indicate regions magnified in the inset. In the inset, the dotted white line outlines distal branches of virus spread. The high level of HIV RNA expression appears yellow. Nuclei were counterstained with hematoxylin. (**B**) Co-localization analysis of HIV RNA and p24 by combined multiplex fluorescent RNAscope ISH and IHC. The spatial overlap between HIV RNA (red) and p24 (green) results in a yellow color. Nuclei were labeled with DAPI (blue). In the merged panel, the dotted white line outlines distal branches of virus spread detected as red dots of a single copy of viral RNA. (**C**) Representative image of HIV RNA^+^hCD3^+^ T cell (yellow arrow) surrounded by hCD163^+^ macrophages in the spleen of the HIV-infected untreated NSG-BLT mouse. (**D**) HIV RNA^+^hCD163^+^ macrophages (yellow arrow) in the spleen of EFdA-suppressed mice occasionally exhibited a low level of p24 expression. (**E**) The yellow arrow indicates an HIV RNA^+^hCD163^+^ macrophage abundantly expressing p24 in the spleen of NSG-BLT mouse two weeks after EFdA cessation. For the overlay images shown on the left in panels (**C**,**D**), and (**E**), separate overlay panels are shown. Representative images are shown of spleens from untreated mouse #37 (**C**), EFdA-suppressed mouse #6 (**D**), and mouse #44 2 weeks after EFdA cessation (**E**). The results (**C**–**E**) are representative of those observed in at least 3 multiplex fluorescent RNAscope/IHC experiments using spleens from 5 or 6 mice per group.

**Figure 8 viruses-11-00256-f008:**
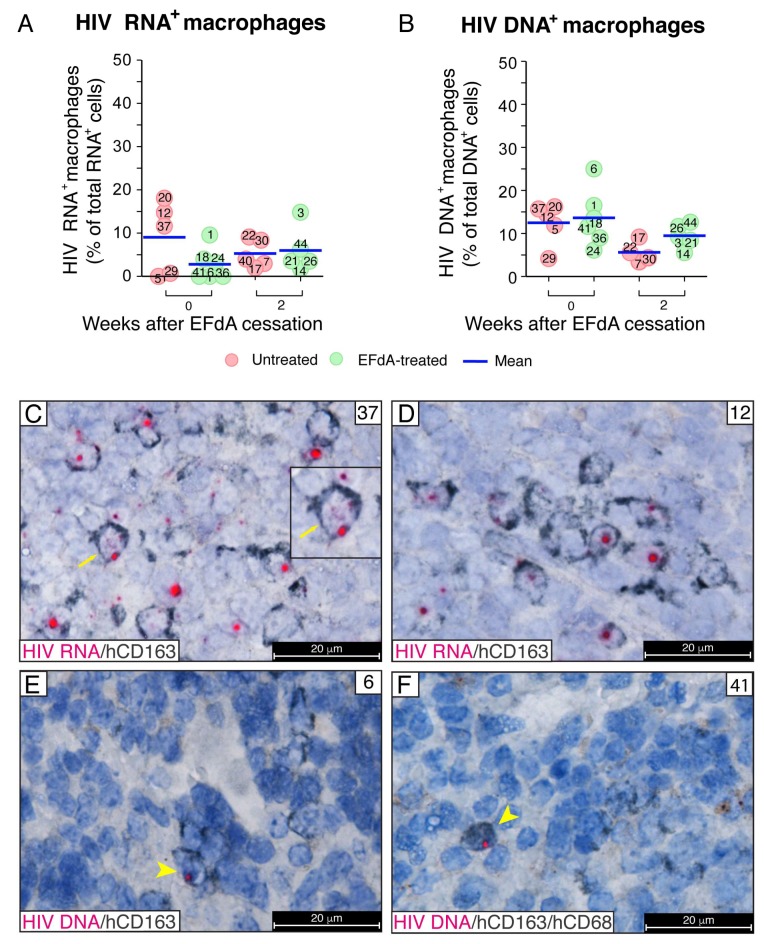
Human CD163^+^ macrophages harbor HIV proviruses during fully suppressive EFdA treatment. HIV-infected macrophages in NSG-BLT mouse spleens were detected by combined chromogenic RNAscope ISH and immunohistochemistry using anti-human CD163 mAb. HIV RNA was identified using an HIV-*gag-pol* antisense probe (**A**, **C**, **D**), and HIV DNA was identified using an HIV-1 clade B sense probe (**B**, **E**, **F**). Thirty to fifty images of representative white pulp regions containing human lymphoid and myeloid cells were acquired at a magnification of 630×. ImageJ software enumerated HIV RNA^+^ cells, HIV DNA^+^ cells, and hCD163^+^ cells. The numbers of HIV RNA^+^ macrophages (**A**) and HIV DNA^+^ macrophages (**B**) were normalized to total HIV RNA^+^ or DNA^+^ cells per image and expressed as a percentage. The Mann–Whitney U-test performed the statistical comparison between groups and were not significant (*P* > 0.05). HIV RNA (**C**,**D**) and HIV DNA (**E**,**F**) were visualized by alkaline phosphatase-Fast Red fluorescence detection. (**C**–**F**) Nuclei were counterstained with hematoxylin. The yellow arrow shows hCD163^+^ macrophages containing HIV RNA signals in the nucleus (because the probe also detects the sense strand of integrated HIV DNA) and cytoplasm. The yellow arrowhead shows HIV DNA^+^ macrophages identified by anti-human CD163 mAb (**E**) or a pool of anti-human CD163 and CD68 mAbs (**F**). Tissue numbers in (**C**–**F**) indicate mouse numbers shown in panels (**A**) and (**B**).

**Figure 9 viruses-11-00256-f009:**
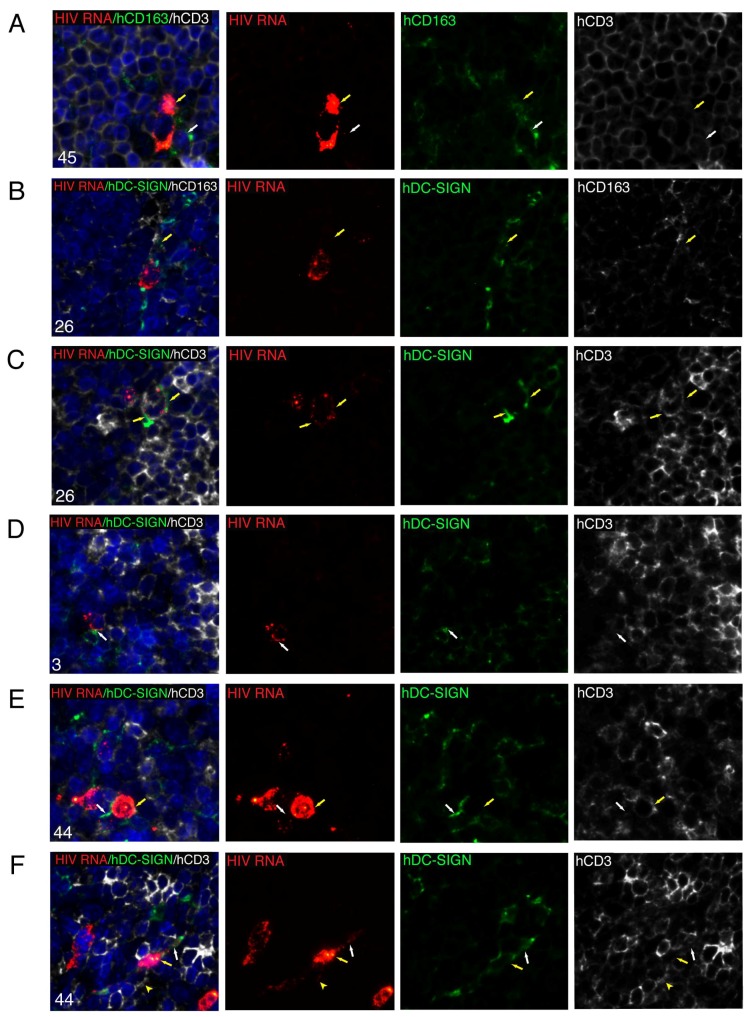
Tracking HIV spread in NSG-BLT mouse spleens 2 weeks after EFdA cessation. Multiplex fluorescent RNAscope ISH for HIV RNA and IHC for human (**A**) CD163 and CD3, (**B**) DC-SIGN and CD163, and (**C**–**F**) DC-SIGN and CD3 performed with spleens from HIV-inoculated NSG-BLT mice before EFdA treatment (**A**) and 2 weeks after drug cessation (**B**–**F**). The results are representative of those observed in at least 3 multiplex fluorescent RNAscope/IHC experiments with spleens from 5 or 6 mice per group. Nuclei were labeled with DAPI (blue). For the overlay images labeled with the mouse number, separate overlay panels are shown. The yellow arrow in A shows HIV RNA^+^ T cell, and the white arrow shows an adjacent hCD163^+^ macrophage containing an HIV DNA signal in the nucleus. The yellow arrow in B indicates a human DC-SIGN^+^CD163^+^ macrophage neighboring an HIV-infected presumptive T cell. (**C**) DC-SIGN expression by adjacent presumptive macrophages is clustered along HIV RNA accumulation sites in a CD3^+^ T cell (yellow arrows). (**D**) The formation of site contact between an HIV RNA^+^CD3^+^ T cell and a DC-SIGN^+^CD3^–^ presumptive macrophage. The spatial overlap between HIV RNA (red) and DC-SIGN (green) at the surface contact point results in a yellow color (white arrow). (**E**) Possible virus spread from an hCD3^+^ T cell abundantly expressing HIV RNA (yellow arrow) to a DC-SIGN^+^ presumptive macrophage containing an HIV RNA signal in the nucleus (white arrow). (**F**) The main body of a DC-SIGN^+^ presumptive macrophage is packed with HIV RNA (yellow arrow), and its periphery contains DC-SIGN clusters and the red punctum of a single HIV RNA molecule (white arrow). The yellow arrowhead shows adjacent CD3^+^ T cells containing several red puncta of HIV RNA. Images were acquired at a magnification of 630×.

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
