# Peer review of "Cellular HIV Reservoirs and Viral Rebound from the Lymphoid Compartments of 4′-Ethynyl-2-Fluoro-2′-Deoxyadenosine (EFdA)-Suppressed Humanized Mice"

_viruses, 2019, doi:10.3390/v11030256_

Round 1

Reviewer 1 Report

The manuscript by Ekaterina Maidji et all., “Cellular HIV reservoirs and viral rebound from the 3 lymphoid compartments of 4’-ethynyl-2-fluoro-2’-4 deoxyadenosine (EFdA)-suppressed humanized mice” addresses the important steps in characterization of HIV-1 lymphoid reservoirs and mechanisms of the spread of infection. By using a combination of RNAscope, in situ hybridization (ISH), and immunohistochemistry (IHC), authors quantitatively investigated the distribution of HIV in the lymphoid tissues of humanized mice during active infection, EFdA suppression, and after drug cessation. They showed that T cell infection is possible in vivo by macrophages at the rebound state via DS-SIGN expressing synapses. Presented data are well illustrated and are novel in humanized mice research.

Minor concerns:

1. Dako company does not exist any more and the clone information for antibodies need to be shown for all used antibodies.

2. Figure 1 contains only human gut. BLT-NSG mice should have gut tissues repopulated by human cells according to other publications. Could authors include gut panels or state that they did not see significant gut repopulation by human cells as their protocol did not include irradiation of mice?

3. The comparison of mouse lymphoid tissues and human GALT for HIV infection is not correct – lines 228-230.

4. Interesting result with mouse #1 was shown on Figure 3. Viral load in plasma was not suppressed but tissue levels significantly reduced. Why is the virus present in the blood during suppression phase? Is it a measurement mistake?

5. Figure 3 panel F – RNAscope shows macrophage membrane associated granules, but not intracellularly. It could be interpreted as attachment of viral particles derived from other infected T cells.

6. Line 307 – where is insert showing HIV DNA in macrophages? Line 309 – where is HIV DNA in T cells? Tissue panels in figure 3 have only HIV RNA. Please include DNA illustration. Might be it was shown on Figure 5?

7. Line 317 – how virus can spread away if cytoplasm does not have HIV RNA copies? Misinterpretation? In macrophages virus bud in multivesicular bodies and cytoplasm should have a lot of viral particles. For me images 3D and 3F looks like “absorption” of particles om macrophage surface.

8. Mouse #1 spleen data on figures 3 and 4 are different. Count of HIV particles on figure 4A more correspond to viral load (not suppressed).

9. Macrophages images with intracellular HIV RNA (Figure 4) confirm that these cells produce virus and two weeks of the treatment was not able to suppress viral replication in macrophages. In the absence of high levels of HIV production by T cells, macrophages do have virus intracellularly in spleen.

10. The question about replication competent virus in suppressed animals is artificial. All mice rebounded and do have replication competent virus.

11. Estimation of viral reservoir by IUPM should be included in Material and Methods section. It is unclear why there are such low differences between non-treated animals who have a lot of cells producing virus and suppressed mice that have a limited number of infectious particles.

Author Response

Response to Reviewer 1 Comments

Point 1: Dako company does not exist any more and the clone information for antibodies need to be shown for all used antibodies.

Response 1: The following text has been modified to remove Dako and to include clone information in lines 113-117 and 159-162: 

“Anti-human CD45 (clone H130, BD Biosciences) and anti-mouse CD45 (clone 30-F11, BD Biosciences) antibodies were used to differentiate mouse from human leukocytes. Human CD45+cells were phenotyped using antibodies specific for human CD3 (clone UCHT1, Beckman Coulter), CD4 (clone RPA-T4, Biolegend), CD14 (clone TüK4), and CD19 (clone HIB19, Biolegend).”

“Primary antibodies were mouse mAb anti-HIV-1 p24 (183-H12-5C) from the NIH AIDS Reagent Program and anti-human CD3 (clone F7.2.38, Diagnostic BioSystems), CD163, (Leica Biosystem), CD68 (clone KP-1, Agilent), rabbit mAb anti-human CD163 (EPR14643-36, Abcam) and CD3 (SP7, Abcam), and rabbit polyclonal Ab anti-human DC-SIGN (Abcam).”

Point 2: Figure 1 contains only human gut. BLT-NSG mice should have gut tissues repopulated by human cells according to other publications. Could authors include gut panels or state that they did not see significant gut repopulation by human cells as their protocol did not include irradiation of mice?

Response 2: We thank the Reviewer for this excellent point suggesting we investigate the distribution of HIV RNA+cells in the gut of HIV-inoculated BLT mice with and without irradiation. We hope to explore this in future studies. In this manuscript, the GALT regions of rectosigmoid biopsies from untreated HIV-infected individuals(Fig. 1G & H) were used exclusively as a positive control for HIV detection. The text was modified accordingly in lines 251-253. 

“GALT regions of rectosigmoid biopsies from untreated HIV-infected individuals [49]were used as a positive control for HIV detection (Figure 1G, H)”

Point 3: The comparison of mouse lymphoid tissues and human GALT for HIV infection is not correct – lines 228-230.

Response 3:This comparison has been removed (lines 251-253). 

Point 4: Interesting result with mouse #1 was shown on Figure 3. Viral load in plasma was not suppressed but tissue levels significantly reduced. Why is the virus present in the blood during suppression phase? Is it a measurement mistake?

Response 4: We acknowledge that EFdA-treated mouse #1 had a high viral load in the plasma comparable to the untreated mice, whereas HIV RNA cells in the spleen were significantly lower than in most untreated mice. This may be because circulating HIV-infected cells in the peripheral blood of this mouse may not have died as quickly during EFdA suppression as those in the other treated mice. Please note that mouse #1 also had substantially more HIV RNA+cells in the spleen than all the other treated mice, which provides evidence of a correlation between viral load and tissue level rather than a measurement mistake.

Point 5: Figure 3 panel F – RNAscope shows macrophage membrane associated granules, but not intracellularly. It could be interpreted as attachment of viral particles derived from other infected T cells.

Response 5: We agree that the interpretation of the discrete puncta of single HIV RNA molecules detected in cells adjacent to the highly infected cell would be more accurate using a multiplex fluorescent RNAscope/IHC assay (Figure 7 and 9). Therefore, we removed the Figure 3F interpretation in line 343.

Point 6: Line 307 – where is insert showing HIV DNA in macrophages? Line 309 – where is HIV DNA in T cells? Tissue panels in figure 3 have only HIV RNA. Please include DNA illustration. Might be it was shown on Figure 5?

Response 6: We apologize for the error in lines 307 and 309. It has now been corrected in lines 336 and 338. We acknowledge that the tissue panel in Figure 3D illustrates HIV RNA. However, the text says “CD3+T cells with a bright fuchsia dot in the nucleus indicating detection of the sense strand of integrated HIV DNA (Figure 3D)”. 

We provide this explanation for the phenomenon in lines 212-214: “Although the RNAscope gag-pol probe that was used in the assay is optimally designed to detect HIV RNA, it may also detect the sense strand of integrated HIV DNA”.

An illustration of HIV RNA and DNA in macrophages is shown in Figure 8C-F.

Point 7: Line 317 – how virus can spread away if cytoplasm does not have HIV RNA copies? Misinterpretation? In macrophages virus bud in multivesicular bodies and cytoplasm should have a lot of viral particles. For me images 3D and 3F looks like “absorption” of particles om macrophage surface.

Response 7: We have removed this interpretation from line 343.

Point 8: Mouse #1 spleen data on figures 3 and 4 are different. Count of HIV particles on figure 4A more correspond to viral load (not suppressed).

Response 8: We acknowledge that spleen data for mouse #1 are different in Figures 3 and 4. This discrepancy has two explanations: 1) In Figure 3, HIV RNA+cells were represented as cells per mm2whereas in Figure 4, the number of HIV RNA+cells was calculated per 10cells because comparisons were performed between different organs (spleen, lung, and Thy/Liv implants). This is described in the Materials and Methods in lines 182-190. 2) Figures 3 and 4 represent data from two RNAscope/IHC assays using two different cross-sections of mouse #1 spleen taken within a distance of 20 mm.

Point 9: Macrophages images with intracellular HIV RNA (Figure 4) confirm that these cells produce virus and two weeks of the treatment was not able to suppress viral replication in macrophages. In the absence of high levels of HIV production by T cells, macrophages do have virus intracellularly in spleen.

Response 9: We thank the Reviewer for this excellent point. Indeed, our study was focused on macrophages as an intriguing source of HIV persistence in the lymphoid tissues of HIV-infected BLT mice during EFdA suppression.

Point 10: The question about replication competent virus in suppressed animals is artificial. All mice rebounded and do have replication competent virus.

Response 10:We agree that this point was not well explained and have modified the text to include the following in lines 408-410:

Next, we verified that cells harboring HIV DNA during EFdA suppression contain replication-competent virus.”

Point 11: Estimation of viral reservoir by IUPM should be included in Material and Methods section. It is unclear why there are such low differences between non-treated animals who have a lot of cells producing virus and suppressed mice that have a limited number of infectious particles.

Response 11:The methods for the IUPM assay are included in Materials and Methods section 2.4 (HIV inoculation and EFdA treatment), lines 136-139.

The IUPM assay was performed with 32,000, 10,000, and 3,200 spleen cells cocultivated in duplicate with normal donor PBMC and p24 detection by ELISA at day 7. The upper limit of detection was therefore 3.75 log10 infectious units per million spleen cells and was likely greater than this for 2 of the untreated mice. We indicate in the legend for Figure 5 that these maximum values are >3.75 log10 IUPM. Please note that while two of 5 untreated mice had >3.75 log10 IUPM spleen cells, 0 of 6 treated mice had this maximum value. We now state this in lines 431-435.

Reviewer 2 Report

The goal of the study reported by Maidji et al. was to investigate, at the cellular level, the formation and persistence of virus reservoirs in lymphoid tissues of HIV-1-infected humanized mice. Using the NSG-BLT mouse model, the authors developed an experimental infection and antiviral protocol of HIV persistence in which virus replication was monitored in lymphoid compartments first during active infection, then after virus suppression through treatment with a reverse-transcriptase inhibitor (EFdA), and finally after drug cessation and viral rebound. Using a combination of RNAscope in situ hybridization and immunohistochemistry techniques, the authors were thus able to detect and visualize infected cells, including T cells but also tissue macrophages, in several lymphoid compartments from spleen, lung, gut and the thy/Liv implant, even after virus suppression through antiviral treatment. The originality of this work is related, first, to the characterization of infected cells in lymphoid compartments such as lungs where in vivo virus replication was not  really explored before,  and second, to the special focus on detection of infected macrophages in lymphoid tissues.

While the specific aims and the experimental strategy used in this report are relatively well defined, and the conclusions are generally supported by the data presented in the manuscript, the main concern is related to the overinterpretation of the data reported in the last figure of the manuscript (Figure 9), in which the authors claimed that there were able to track HIV dissemination from infected T cells to macrophage targets.  Because of the poor quality of the images shown in this figure regarding tissue staining with anti-DC-SIGN and anti-CD3, I think it is not possible to discriminate between virus-donor cells and target cells from these images. In some images, it seems that some T cells could be also stained for DC-SIGN, and reciprocally macrophages could be positive for CD3 staining. In addition, it seems that there is a confusion in the Discussion Section (pages 25 and 26) about these results regarding the mechanisms of either trans infection of target T cells from virus-donor macrophages or trans infection of target macrophages from virus-donor T cells. To my knowledge, there are only 2 reports in the literature from the respective groups of Q. Sattentau (Baxter, et al, 2014) and S. Benichou, regarding the mechanisms of cell-to-cell viral infection of macrophages targets from infected T cells. While Baxter et al (2014) showed that macrophage target were infected through phagocytosis of infected T cells, Bracq et al. have reported that virus cell-to-cell transfer from infected T cells toward macrophage targets was related to cell-fusion leading to the formation of multinucleated giant cells (MGCs). These in vitro observations could be discuss in the manuscript.

In conclusion, I think the authors should remove or reinterpret the results reported in the figure 9 without concluding several times in the manuscript (Abstract, Introduction, Results and Discussion Sections) that they are related to virus spreading from infected T cells to macrophage targets. In addition, the mechanisms reported in the litterature for trans infection of either target T cells or macrophages could be further discussed.

Other minor (but important) point, there are a lot of mistakes on pages 7, 9 and 10 of the Results Section regarding the numbering of Figures 1, 2 and 3.

Author Response

Response to Reviewer 2 Comments

Point 1: While the specific aims and the experimental strategy used in this report are relatively well defined, and the conclusions are generally supported by the data presented in the manuscript, the main concern is related to the overinterpretation of the data reported in the last figure of the manuscript (Figure 9), in which the authors claimed that there were able to track HIV dissemination from infected T cells to macrophage targets. 

Response 1: We agree with the Reviewer that it is very difficult to discriminate in situbetween virus-donor cells and target cells. In response to this important point,we reinterpreted the results in Figure 9 in lines 561-575 and modified all conclusions and discussions about HIV dissemination from infected T cells to macrophage targets in lines 25-28, 622-625, 772-778, and 802-806.

Abstract:“Our data provided the visualization and direct measurement of the early steps of HIV reservoir expansion within anatomically intact lymphoid tissues soon after EFdA cessation and suggest a strategy to enhance therapeutic approaches aimed at eliminating the HIV reservoir.”

Results:“Detailed tissue analyses allowed us to monitor the following steps of HIV spread in vivo after EFdA cessation and viral rebound: 1) HIV RNA accumulation along the CD3+cell membrane of T cells was often accompanied by DC-SIGN clustering on the cell surface of adjacent presumptive macrophages, resembling DC-SIGN expression at the virological synapse [54-56](Figure 9C, yellow arrows). 2) At the site of surface contact between a hCD3+T cell (white) and a DC-SIGN+presumptive macrophage, HIV virions occasionally appeared with a yellow color that results from the spatial overlap of a red HIV RNA punctum and green DC-SIGN protein, resembling virion transport across the virological synapse [57, 58](Figure 9D). 3) DC-SIGN was highly expressed at the site of surface contact between a HIV-infected T cell abundantly expressing viral RNA and adjacent macrophages with evidence of initial HIV infection, suggesting HIV spread (Figure 9E).”

“Together, these results demonstrate that the spleen of HIV-inoculated NSG-BLT mice is enriched with CD163+macrophages and that these macrophages are capable of harboring HIV during EFdA suppression. Furthermore, during the 2 weeks after EFdA cessation, they form numerous surface site contacts with T cells expressing DC-SIGN that seem to be involved in viral spread.”

Discussion:“The monitoring of HIV spread in the NSG-BLT mouse spleen 2 weeks after EFdA cessation further suggests that highly expressed DC-SIGN on macrophages at the site of cell contact colocalizing with concentrated HIV RNA puncta could be involved in the formation of virological synapses, thus facilitating efficient viral spread and rapid viral rebound.” 

Point 2: To my knowledge, there are only 2 reports in the literature from the respective groups of Q. Sattentau (Baxter, et al, 2014) and S. Benichou, regarding the mechanisms of cell-to-cell viral infection of macrophages targets from infected T cells. … These in vitro observations could be discuss in the manuscript.

Response 2: We thank the Reviewer for this important suggestion. The following text was added into the Discussion in lines 759-762.

“Macrophages participate in HIV dissemination by various mechanisms of cell-to-cell transfer including selective capture and engulfment of HIV-infected T cells leading to efficient infection of the macrophage [91], heterotypic envelope-dependent cell fusion leading to the formation of lymphocyte-macrophage fused cells [92], and virological synapse formation [75, 76, 93, 94].”

Point 3: Other minor (but important) point, there are a lot of mistakes on pages 7, 9 and 10 of the Results Section regarding the numbering of Figures 1, 2 and 3.

Response 3: We apologize for the error in the numbering of the figures. This has now been corrected in lines 251, 330, 336, 338, 342, 343, 478, and 686.